

# Global patterns of Earth's dynamic topography since the Jurassic

Michael Rubey[1], Sascha Brune[2], Christian Heine[3], David Rhodri Davies[4], Simon E. Williams[1], Ralph Dietmar Müller[1]

[1]Earthbyte Group, School of Geosciences, The University of Sydney, Australia
[2]Helmholtz Centre Potsdam, GFZ German Research Centre for Geosciences, Potsdam, Germany
[3]Shell International Exploration & Production B.V., The Hague, Netherlands
[4]Research School of Earth Sciences, The Australian National University, Canberra, Australia

*Correspondence to*: M. Rubey (michael.rubey@sydney.edu.au)

**Abstract.** We evaluate the spatial and temporal evolution of Earth's long-wavelength surface dynamic topography since the
Jurassic, using a series of high-resolution global mantle convection models. These models are Earth-like in terms of
convective vigour, thermal structure, surface heat-flux and the geographic distribution of heterogeneity. The models generate
a degree-2 dominated spectrum of dynamic topography, with negative amplitudes above subducted slabs (i.e. circum-Pacific
regions and southern Eurasia) and positive amplitudes elsewhere (i.e. Africa, north-west Eurasia and the central Pacific).
Model predictions are compared with published observations and subsidence patterns from well data, both globally and for
the Australian and South African regions. We find that our models reproduce the long-wavelength component of these
observations, although observed smaller-scale variations are not reproduced. We subsequently define "geodynamic rules" for
how different surface tectonic settings are affected by mantle processes: (i) locations in the vicinity of a subduction
zone show large negative dynamic topography amplitudes; (ii) regions far away from convergent margins feature long-term
positive dynamic topography; (iii) rapid variations in dynamic support occur along the margins of overriding plates (e.g.
Western US) and at points located on a plate that rapidly approaches a subduction zone (e.g. India and Arabia). Our models
provide a predictive quantitative framework linking mantle convection with plate tectonics and sedimentary basin evolution,
thus improving our understanding of how subduction and mantle convection affect the spatio-temporal evolution of basin
architecture.

## 1 Introduction

At short spatial scales, Earth's topography depends on interactions between crustal structures, tectonic deformation, and
surface processes. However, at length-scales exceeding the flexural strength of the lithosphere, topography is an expression
of the force-balance acting at Earth's free surface. This balance has: (i) an *isostatic* component, which depends on the
composition and thickness of the crust and lithospheric mantle; and (ii) a *dynamic* component, which is the deflection of
Earth's surface resulting from vertical stresses generated by convection in the underlying mantle. Their relative importance
varies according to the tectonic setting. Most orogens show patterns of thickened continental crust and hence, isostatically





compensated topography (e.g. Frisch et al., 2011), while the anomalous elevation of the stable South African craton has been attributed to dynamic topography generated by deep-mantle upwellings (e.g. Lithgow-Bertelloni and Silver, 1998; Gurnis et al., 2000).

Dynamic topography generates significant surface deflections in both continental and oceanic regions and influences the
shape of the ocean surface via its effect on the geoid (e.g. Ricard et al., 2006). The connection to sea level and continental flooding is direct: as continents migrate over mantle upwellings (regions of positive dynamic topography) and downwellings (negative dynamic topography), vertical motions may be sufficiently large to control the emergence and submergence of continents (e.g. Mitrovica et al., 1989), thus generating sedimentary basins or erosional surfaces, respectively. Dynamic topography is identified in Earth's present-day relief via deviations from isostasy, yielding the so-called residual topography
field (e.g. Kaban et al., 1999; Doin and Fleitout, 2000; Crosby et al., 2006; Winterbourne et al. 2009; Flament et al., 2013; Hoggard et al., 2016). It is recorded via anomalous vertical motion events in the basin infill (e.g. Mitrovica, 1989, Heine et al., 2008) or by long-wavelength uplift in non-basinal areas, as seen in plateau exhumation and geomorphic features (e.g. Roberts & White, 2010). It is, therefore, a fundamental component of Earth's topographic expression and an improved understanding of the history of Earth's surface topography can only be achieved by successfully accounting for its spatial
and temporal variations.

There are two principal ways to retrieve the component of dynamic support: (i) the "top-down", inverse approach, which is based on geophysical and geological observations and subtracts isostatic topography from observed surface elevations; and (ii) the "bottom-up" strategy, which infers dynamic support by determining mantle flow from Earth's internal buoyancy and viscosity distribution. In this study, we follow the bottom-up strategy and discuss our findings in relation to results from the
top-down approach (e.g. Panasyuk and Hager, 2000; Kaban et al., 2003; Kaban et al., 2004; Steinberger, 2007; Heine et al., 2010; Flament et al., 2013; Hoggard et al. 2016). Discrepancies between these two basic strategies lead to much discussion in the literature (e.g. Flament et al., 2013; Molnar et al. 2015; Colli et al., 2016; Hoggard et al., 2016). Molnar et al. (2015) argue that only small free-air gravity anomalies (<30-50 mGal) are caused by long-wavelength topography, interpreting most of the signal as isostatic. They suggest a maximum of 300 m of dynamic support is indicated by long-wavelength gravity
anomalies, which is in contrast to many published geodynamic models (e.g. Flament et al., 2013). However, Colli et al. (2016) demonstrate that mantle convection with a depth-dependent viscosity generates small perturbations (geoid: 37 m, gravity: 8 mGal), considerable dynamic topography (1.1 km), and a gravitational admittance mostly less than 10 mGal/km. This supports the idea that relatively small long-wavelength free-air gravity anomalies do not preclude dynamic topography at amplitudes substantially larger than 300 m, and helps to underpin the results of Hoggard et al. (2016) of variations of 1 km
of dynamically induced topography, inferred from observations.

Numerous bottom-up studies have been undertaken over recent years in an attempt to constrain the amplitude, spatial pattern and time-dependence of dynamic topography using numerical modelling. These studies examined the temporal evolution of dynamic support by forward modelling (e.g. Ricard et al., 1993; Zhang et al., 2012; Flament et al., 2013), backward advection (e.g. Conrad and Gurnis, 2003; Moucha et al., 2008; Heine et al., 2010) or more advanced adjoint schemes (e.g.



Bunge et al., 2003; Liu et al., 2008; Spasojevic et al., 2009; Glišović and Forte, 2017). Present-day dynamic topography is discussed on a global scale by Ricard et al. (1993), Steinberger (2007), Zhang et al. (2012) and Flament et al. (2013), but relatively few studies (e.g. Zhang et al. (2012), Flament et al. (2013)), which use the forward modelling approach, have examined the temporal evolution of dynamic support globally.

In this paper, we also use numerical forward modeling to compute the transient evolution of dynamic topography. The distribution of heterogeneity in our models is generated by solving the equations relevant to mantle convection inside a global spherical shell, with plate velocities, derived from a global plate kinematic model (Seton et al., 2012, with modifications by Shephard et al., 2013), featuring topological plate boundaries extending back to 200 Ma, imposed as a surface boundary condition. We compare our predicted present-day dynamic topography to previous numerical studies

before discussing the temporal evolution of dynamic topography within selected regions. Finally, we propose a categorisation of regions on Earth's surface in terms of their exposure to effects of dynamic support, which is corroborated by an automated classification technique.

## 2 Numerical Model

### 2.1 Model setup

We use the global mantle convection code TERRA (e.g. Baumgardner, 1985; Bunge et al., 1997; Davies and Davies, 2009; Wolstencroft et al., 2009; Davies et al., 2013; Wolstencroft and Davies, 2017), which solves the Stokes and energy equations relevant to mantle convection inside a spherical shell, with appropriate material properties and boundary conditions (Fig. 1). Calculations are performed on a mesh with ~80 million nodal points, thus providing the resolution (~22.5 km nodal spacing) necessary to explore mantle flow at Earth-like convective vigour. Our models incorporate compressibility, via the Anelastic

Liquid Approximation (ALA), with radial reference values represented through a Murnaghan equation of state (Murnaghan, 1944). Isothermal boundary conditions are specified at the surface (273K) and CMB (Model M1 and M3 - 4000 K; Model M2 - 3500 K), with the mantle also heated internally, at roughly chondritic rates. A free-slip boundary condition is specified at the CMB, whilst surface velocities are assimilated via 200 Myr of plate motion histories (Seton et al. 2012; Shephard et al. 2013), generated by the plate tectonic reconstruction software GPlates (Boyden et al., 2011; Fig. 1), at discrete 1 Myr

intervals. In our reference model (M1), we prescribe a radial viscosity profile with four layers, namely lithosphere ($10^{23}$ Pa s), sub-lithospheric upper mantle ($10^{20}$ Pa s), transition zone ($10^{21}$ Pa s), and lower mantle ($10^{23}$ Pa s). Viscosity also varies with temperature $T$, following the relation:

$$\eta = \eta_{ref}(r)e^{4.61(0.5-T^*)}, \text{ with } T^* = \frac{T-T_{surf}}{T_{CMB}-T_{surf}},$$

where $T^*$ represents the non-dimensionalised temperature. For initial conditions, we approximate the unknown Jurassic

mantle heterogeneity structure by applying a 70 Myr long pre-conditioning stage, where global plate configurations are fixed to the oldest available reconstruction at 200 Ma. From here, models run forwards, for 200 Myr, towards the present-day. Key

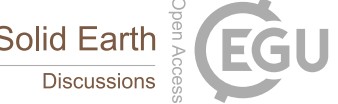



model parameters are listed in Table 1, with reference values also illustrated in Fig. 2. Three separate cases are examined with the variations between these cases summarised in Table 2.

Dynamic topography is computed from normal stresses at the model surface using the following formula:

$$h = \frac{1}{\Delta\rho g}\sigma_{rr}\frac{\eta_0}{\eta_{ref}},$$

where $\sigma_{rr}$ is the normal stress, $\Delta\rho$ the density difference between surface and air, and $g$ the acceleration due to gravity. $\eta_0$ is an estimate of the asthenospheric viscosity ($3\times10^{19}$ Pa s; e.g. Mitrovica and Forte, 2004; Steinberger and Calderwood, 2006; Petersen et al., 2010), while $\eta_{ref}$ is the models' reference viscosity. Note that to calculate dynamic topography, we want to extract mantle convection related normal stresses that are independent from lateral buoyancy variations within the lithosphere, such as generated at the margins of continents or within the cooling oceanic lithosphere. As such, we replace the

model's thermal structure above 225 km depth with a constant mantle temperature of 2150 K. We subsequently re-run the model for one time step and use the resulting surface normal stresses to evaluate dynamic topography. Similar approaches to discard the lithospheric buoyancy structure from the dynamic topography signal have been applied by Lithgow-Bertelloni and Silver (1998), Steinberger (2007), Conrad and Husson (2009) and Flament et al. (2013) for structures above 325, 220, 300, and 350 km depth, respectively.

**2.2 Model robustness**

To assess the robustness of our results, we vary two key parameters of our reference model, M1, and compare the temporal evolution of dynamic topography at selected locations. These synthetic sample points have been picked to illustrate the effect on regions where either the influence of dynamic support at present day or through time is debated, or whose geological setting prompts that question. Model M2 employs the same viscosity profile as M1 (cf Fig. 2 and Table 2) but differs in

CMB temperature, where a temperature of 3500 K is prescribed instead of 4000 K. Model M3, on the other hand, utilises the same CMB temperature of 4000 K, but its viscosity profile differs from model M1: lower mantle viscosity is reduced from 1 x $10^{23}$ Pa s to 3 x $10^{22}$ Pa s.

The computed dynamic topography is not strongly affected by these variations, as demonstrated by the elevation histories of chosen sample locations in Fig. 3. Decreasing the CMB temperature has a negligible effect, with models M1 and M2

predicting very similar dynamic topography amplitudes and trends. Decreasing the lower mantle viscosity has a more pronounced influence, as slab penetration into the lower mantle is facilitated and slab-sinking rates are increased. This has two principal effects: (i) due to viscous coupling, faster sinking slabs increase the amplitude of negative dynamic topography; and (ii) the suction effect of a faster sinking, detached slab fades quicker after subduction slows down or ceases. Nevertheless, the first-order characteristics of all three models (Fig. 3) are in good agreement, indicating that long-

wavelength dynamic topography is principally controlled by the plate tectonic history and the location of subduction zones. Accordingly, for the remainder of this paper, we focus solely on our reference model, M1.





## 3 Model Results and Implications

### 3.1 Predicted mantle structure

Snapshots of the mantle structure predicted in model M1 are presented in Fig. 1, illustrating that the upper mantle planform is dominated by strong downwellings in regions of present-day plate convergence. In the mid-mantle, cold downwellings are

prominent beneath North America (the subducted Farallon slab) and southern Eurasia (the former Neo-Tethys ocean), whilst remnants of older subduction (those encircling the former super-continent Pangea) are visible above the CMB. In our model M1, slabs sink at rates of 1.5 cm/yr, comparable to those inferred from seismic tomography (e.g. Li. et al. 2008; Simmons et al. 2010). These downwellings modulate the location of upwelling flow, such that it becomes concentrated beneath Africa and the Pacific (e.g. McNamara & Zhong, 2005; Davies et al. 2012; 2015). The Pacific anomaly is approximately circular,

whilst the African anomaly is a NW-SE trending structure, which to the north curves eastward under Europe and to the south extends into the Indian Ocean. Note that due to: (i) the short evolution times of our models; and (ii) the limited temperature dependence of viscosity, plumes play a negligible role in the convective planform. Accordingly, the regions of upwelling flow beneath Africa and the Pacific discussed above principally represent passive return flow from subduction.

### 3.2 Global and regional dynamic topography

The spatio-temporal evolution of dynamic topography from case M1 is illustrated in Fig. 4A in the mantle reference frame and in Fig. 4B in a plate frame of reference. Note the strong correspondence of the upper mantle thermal heterogeneity distribution with negative dynamic topography in regions of present-day and former subduction, and positive dynamic topography elsewhere. Before we further interpret our results we compare them with previous geodynamic modelling studies, as well as with selected observations on a regional scale: we evaluate our model against well data at Australian locations,

which experienced two marked periods of anomalous subsidence and uplift since the Jurassic. Finally we consider observations that constrain the uplift history of South Africa. Due to its particular geological setting South Africa has been the target of many studies on constraining its uplift history and assessing to what extent and when it experienced dynamic support.

### 3.2.1 Global models of dynamic topography

To place our results in the context of previous studies, we compare our predictions of present-day dynamic topography (Fig. 4A) to those of Ricard et al. (1993), Steinberger (2007), Conrad and Husson (2009), Spasojevic and Gurnis (2012) and Flament (2013) shown in Fig. 5. All models display a dominant degree two pattern, whereby the Pacific part is most prominent in all studies. In Steinberger (2007), Conrad and Husson (2009) and Spasojevic and Gurnis (2012), the African / Indian Ocean region consists of a cluster of upwellings with moderate amplitudes as opposed to a more coherent strong high

in the other models. The reason is that these models infer mantle buoyancy structure by converting seismic tomography to temperature resulting in smaller topography wavelengths. In the other models mantle heterogeneities derive from subduction



history, where upwellings feature less internal heterogeneity. In a very recent study, Guerri et al. (2016) examine combinations of 11 different crustal models (Guerri et al., 2015) with 10 different tomography models after obtaining mantle density through thermodynamic modelling of various chemical compositions of the mineralogical assemblage. This yields a peak-to-peak dynamic topography amplitude exceeding 3 km for all models. Other studies suggest that amplitudes for long-

wavelength ($10^3$ km) dynamic topography should be smaller than ±1 km (Wheeler and White, 2002; Molnar et al., 2015; Hoggard et al., 2016). In contrast to previous models, our model M1 agrees well with this suggestion, mapping deep mantle return flow above the Pacific and African LLSVPs at amplitudes not exceeding ±1 km.

### 3.2.2 Australia

The tectonic stability of the Australian continent and its relatively large displacement relative to the underlying mantle since

the Cretaceous makes it an ideal location for investigating mantle induced topography changes. It is the only continent to have undergone two distinct episodes of dynamic topography during the last 200 Myr: (i) a pronounced west-east dominated warping during the middle and late Jurassic / early Cretaceous (Gurnis et al., 1998) caused by subduction along its eastern margin as the Phoenix plate subducted westwards under East Gondwana, and (ii) a south-north tilt due to subduction of the Australian plate underneath the Eurasian and Pacific plates (Fig. 6) that started in the Miocene and continued through to the

present-day (Sandiford, 2007). The effect of transient, mantle convection-induced vertical Australian motions has been the subject of many observation- and model-based research efforts (Veevers, 1984; DiCaprio et al., 2009; Heine et al., 2010; Czarnota et al., 2014).

To ground-truth our predicted dynamic topography we focus on tectonic uplift / subsidence histories from five selected Australian exploration wells, located in opposite corners of the Australian continent. The well data were retrieved from

Geoscience Australia's PIMS system online (http://dbforms.ga.gov.au/pls/www/npm.pims_web.search) (Fig. 7).

For the well on the North West Shelf we find good agreement of the long-term Cenozoic elevation history (Fig. 7 a; Müller et al., 2000). The curve shows that the area was subjected to gradually increasing subsidence during the Cenozoic. Also, for the remaining three wells (Figs. 7 b, c, d) our modelled vertical motion histories follow the backstripped subsidence patterns generally quite well.

The Gandara 1 well in the north-western part of Australia shows a distinctly different temporal subsidence behaviour from the most eastern one in the Surat Basin, Cecil Plains West 1. We explain this by the evolution of dynamic topography shown in the maps of Fig. 6. The former well is not affected by subducted, cold, dense slab material from the East Gondwana subduction zone. When, from around 70 Ma onwards, Australia starts progressively moving north this area experiences the increasing influence of downward mantle flow related to the subduction of the Australian lithosphere along the Indonesian

subduction zone system. The Cecil Plains well, on the other hand, mostly displays the Jurassic and early Cretaceous East Gondwana subduction setting, while Australia's Cenozoic tilting only has a moderate effect: this location indicates phases of accelerating subsidence until about the latest Jurassic, followed by a phase of non-deposition/erosion and a young phase of gentle subsidence. Our models can reproduce the initial Jurassic phase of subsidence. We attribute the time of the





depositional hiatus as the rebounding of the west-east tilt, while the renewed phase of subsidence seems to be related to the Cenozoic tilting.

For the remaining bore holes, Duyken 1 and Mallabie 1 dynamic topography can be explained along analogous lines of arguments (Fig. 6). Both reflect Cretaceous subsidence, with the former being more pronounced as the subducted material

progressively extends farther west in the north of Australia. A relaxation of subsidence can be seen when that influence fades, just to be affected even more profoundly later when Australia's north is drawn down by the slabs of the Indonesian subduction zone. Keeping in mind that dynamic topography varies only slowly over time, we observe a good correlation between model-predicted vertical motions and reconstructed vertical motions from well data. Hence, these data corroborate to the dynamic topography amplitudes of our study.

### 3.2.3 Southern Africa

Southern Africa's unusually high elevation relative to other continents despite being surrounded by passive margins make it an area of particular interest. Braun et al. (2014) analysed sedimentary flux data, which confirmed a major phase of erosion on the southern African Plateau in the Late Cretaceous. By examining paleoprecipitation indicators across the plateau they ruled out climate variability as the main driver for increased erosional activity. Thus, Braun et al (2014) concluded that the

continent migrated over a source of dynamic topography at that time. Tinker et al. (2008) quantified exhumation using apatite fission track dating. While denudation rates during the Cenozoic indicate that mantle processes did not primarily cause the uplift of the region, the authors suggest that mantle buoyancy during the Cretaceous was responsible for enhanced erosion across southern Africa, which, in turn, generated up to 3 km of exhumation (Tinker et al., 2008). Stanley et al. (2013) performed a detailed thermochronometry study across the southern African Plateau employing (U-Th)/He dating. They

suggest a mid-Cretaceous buoyancy increase in the lithospheric mantle to have caused coeval denudation. Our model predicts dynamic support equivalent to ~ 200 m underneath southern Africa, gradually developing in the Cretaceous period (see Fig. 8). It does not reproduce a post-30 Ma uplift phase as revealed by the inversion of river profiles (Paul et al., 2014), which may reflect the lack of active upwellings in our convection model. However, southern Africa is also an area where our results compare particularly well to other modelling efforts (Zhang et al., 2012; Flament et al., 2014).

### 3.3 Modes of long-term dynamic surface topography evolution

In this section we use our model results to establish a set of "geodynamic patterns" for how different continental regions are affected by underlying mantle processes. We define three categories that are exemplified by selected locations, using two alternative approaches. First, we plot time-series of their dynamic topography and visually classify them into these categories, based upon the overall temporal trends in elevation. Secondly, we apply a *k*-means based cluster analysis to the

time-series of dynamic topography, as an independent means to validate our interpretation and explore the consequences of classifying the time-series into different numbers of groups.



### 3.3.1 Category I - Topographic stable areas

The first category comprises locations in topographic stable regions. These areas experience only positive dynamic support over the 200 Myr time period examined generating stable positive topography of a few hundred meters. As can be seen from Fig. 9, these regions lie on a curved band that stretches from the Eurasian to the African plate and extends into the Somalian

plate, comprising locations in Africa, Madagascar, Western and Northern Europe, and Western Siberia. This category generally consists of regions with no adjacent subduction zone. Where points are located adjacent to a subduction zone (for example the Renish Massif next to the Western Tethyan subduction zones, Fig. 9 h), there is insufficient cold, dense slab material in the mantle to cause negative dynamic topography, as evidenced by a computation of subducted oceanic lithosphere volumes (Fig. 10) for the past 200 Myr: the total volume of subducted material in the western Tethyan / present-

day Mediterranean region is of the order to 10000-15000 $km^3$ per 30 arc-minute grid cell, yielding very low average volume fluxes of around 50-75 $km^3$/Myr.

### 3.3.2 Category II – Dynamically subsiding areas

Locations that fall into the second category experience a long-term, continuous decrease in dynamic support. Examples include East and West India, Arabia, Northeast Brazil, the St. Lawrence Gulf region and Western Australia (Figs. 11A and

11B). These regions subside as they approach sites of active or recent subduction. Note that Australia is a special case where two subduction zones exert a sequential influence (cf. Sect. 3.2.2). Here, first the East Gondwana and, subsequently, the Indonesian subduction zone give rise to a decrease in elevation, as is described above.

### 3.3.3 Category III - Fluctuating areas

Locations in this category (Fig. 12) exhibit a non-monotonic temporal evolution with mostly negative dynamic topography.

This is likely to be caused by two factors: (i) the influence of multiple subduction zones, or (ii) the influence of a single subduction zone that exhibits strongly time-dependent characteristics.

Multiple subduction zones affected the dynamic topography evolution of Australia, as discussed above: one can identify two lows in dynamic subsidence when the particular location is influenced by one of these. The Chukchi Sea (Fig. 12a) is a second example where three local minima in dynamic topography correlate with sequential activity of Arctic subduction

zones: the Koni-Taigonos subduction zone at ~ 200 Ma, the Koyukuk and Nutesyn subduction between 140 Ma and ~ 125 Ma, and the Alaskan subduction zone since ~ 110 Ma (Shephard et al., 2013; see Fig. 4A and corresponding Supplementary Material). A third example is the Central Siberian Plateau (Fig. 12g), which was first affected by the East Gondwana subduction zone (~ 180 Ma) and, subsequently (~ 120 Ma onwards), influenced by the East Asian subduction system.

A key time-dependent parameter of a subduction zone is the subduction velocity. Rapid convergence tends to introduce more

cold, dense material into the mantle, which leads to a more pronounced negative dynamic topography. To isolate the role of subduction velocity, we compare dynamic topography time-series at three different subduction zones with the orthogonal



component of the relative velocity between subducting and overriding plate, at the closest subduction zone (Fig. 13). For instance, the Chukchi Sea location exhibits a significant increase in negative dynamic topography about 0 m at 145 Ma to about -500 m at 120 Ma. This modelled dynamic subsidence phase is followed by a short period of rebound (+150 m over 15 Myrs), before the area is again modelled at experiencing dynamic subsidence of more than 500 m between 110-80 Ma.

Here, we can correlate the modelled trench-normal convergence velocities from our input plate kinematic model (Seton et al. 2012; Shephard et al. 2013) directly with predicted dynamic topography evolution over time. Similarly, accelerating convergence from 200 Ma until ca. 150 Ma at the Western Interior Seaway sample point, corresponds to decreasing dynamic topography. This is followed by a short uplift period due to slow convergence before elevation decreases again, driven by the increased convergence velocity. An analogous situation occurs in the South American Amazonian region, where the general

trend of orthogonal convergence correlates well with the evolution of dynamic topography.

It should be noted that the slabs of our models are generally weaker than slabs on Earth, due to simplistic representation of rheology (we neglect yielding, Peierls Creep and other Non-Newtonian effects, which are likely significant - see Garel et al. (2014) for detailed discussion). As a consequence, upon slab descent, slabs often tear and break, which leads to rapid, localised changes in regional dynamic topography (e.g. Fig. 12 c-e; h-l). Similar tears or breaks occur due to abrupt changes in plate motions. The closer a points lies to a subduction zone where this occurs, the larger will be the associated amplitude

of dynamic topography change.

### 3.3.4 Cluster analyses

In the previous section, we isolated three characteristic groups, which represent distinct trends in vertical motion. This classification was based upon a simultaneous consideration of absolute elevations, changes in elevation and the geodynamic

context. Here we use cluster analysis, based upon the $k$-means algorithm (e.g. Lekic & Romanowicz, 2011), as an independent means to validate our classifications (see Supplementary Material for a detailed description). To this end, we search for clusters of similarity within our dynamic topography time-series, at individual surface points, using a grid of nearly $2 \times 10^5$ equidistantly spaced nodes on the sphere, providing a resolution of ~ 28 km on the surface, sampled at 5 Myr intervals in time. The cluster analysis was applied to the dynamic topography time-series from 150 Ma to present – this

excludes the first 50 Myr of the model run where our dynamic topography predictions will be strongly sensitive to the initial condition.

The $k$-means algorithm partitions a given dataset into $k$ clusters such that the sum of the squares of the deviation from the cluster mean is minimised. The number of clusters k is an input parameter of the algorithm. Note that this approach only evaluates elevations at the particular grid point at a certain time and does not account for relative variations (i.e. uplift or

subsidence), nor does it take into account the history of elevations. For a detailed description of the $k$-means algorithm and result for all number of clusters ($k$) see the supplementary material.

We vary $k$ between 2 and 6, which allows us to highlight a variety of spatio-temporal patterns of dynamic topography. Here we discuss $k$=5, because this is the minimal number of clusters that reflects the underlying geodynamics to a first order (Fig.



14). Cluster 5 can be directly associated with our Category I ("topographic stable regions"), while Cluster 4 is an approximate representation of our Category II regions ("dynamically subsiding areas"). To first order, the map in Fig. 14 mirrors our dynamic topography maps in that the topographic high regions (dark blue areas) are concentrated in a region that stretches from Russia over Europe to Africa, mostly far away from subduction zones. Cluster 4 (light blue) corresponds to

subsiding areas, which approach subduction zones. Also, North India falls into this category (as opposed to $k < 5$, see Supplementary Material), describing the history of the subcontinent, starting out as a part of the highly elevated Pangea supercontinent to the plunging of its northern edge as it collides with Asia (Fig. 11A b, g). Arabia, however, has been determined not to reside in this group, but in the "topographic high" group instead, since on average, for most of the observed time period this location displays positive elevations (Fig. 11A l). The same holds true for the borderline case of

the Cuvier Abyssal Plane (Fig. 11A k). Category III ("Fluctuating areas") disintegrates in three distinct clusters (red, orange and green): Cluster 1 with points that remain at negative dynamic topography throughout the entire model run, and Cluster 2 and 3 with locations that experience an initial phase of long-term dynamic subsidence followed by a subsequent uplift. Higher cluster numbers $k$ serve to further partition our categories even more, yet do not reveal any additional information (see Supplementary Material). To summarise, the $k$-means scheme provides an independent instrument for quantitative

analysis of our model data and we find good correspondence to our classification.

**4 Conclusions**

We have demonstrated that the long-wavelength influence of dynamic topography in different continental regions can be classified via their spatio-temporal vicinity to a subduction zone: (i) locations that are far away from subduction zones are generally stable and dynamically supported - these lie on a curved band stretching from the Eurasian plate to the African

plate and extending into the Somalian plate; (ii) locations that move towards subduction zones dynamically subside over time; and (iii) locations that are being influenced by numerous subduction zones or a subduction zone that shows strong time-dependence (i.e. abrupt changes in plate motions of both the overriding and subducting plate) will display a oscillatory history of dynamic topography. While this classification is based upon geodynamic understanding, it is confirmed through an independent machine learning approach, based upon a $k$-means cluster analysis. As geosciences moves towards more

integrated approaches such as source-to-sink studies, our pattern analysis of dynamic topography provides a useful means to evaluate the dynamic relationships between basin infill and sediment sourcing regions to reconstruct regional, spatio-temporal infill histories which are relevant to unravel Earth system dynamics and resource exploration.

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





29      **Figures**

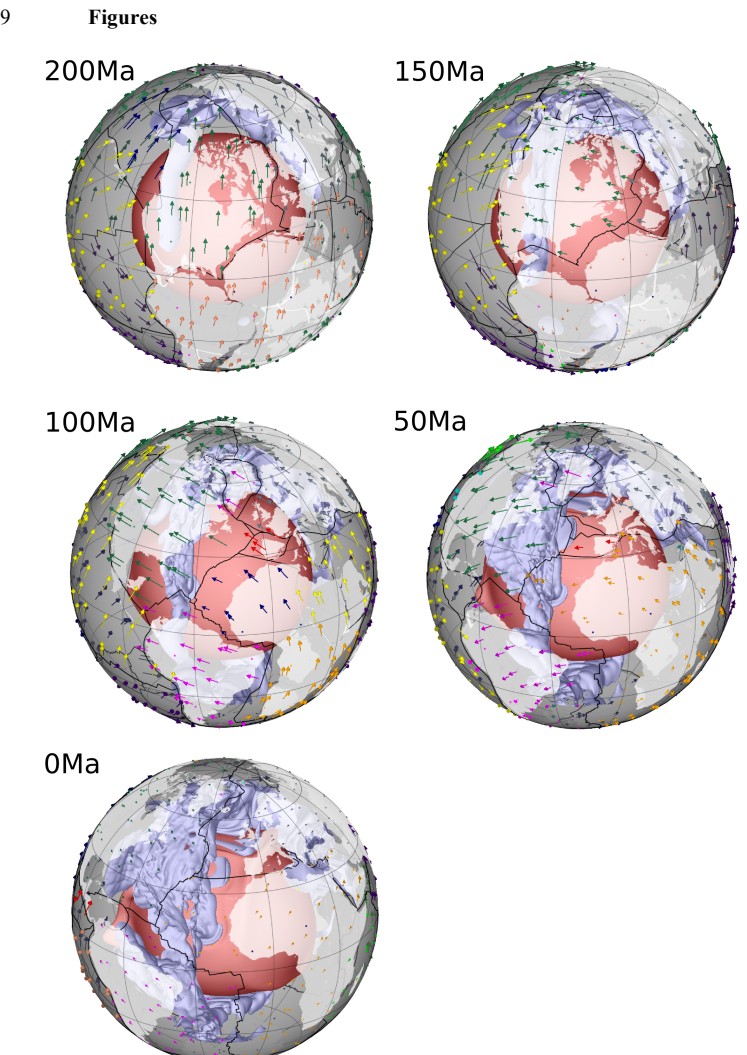

**Figure 1: Schematic depiction of model run showing the evolution of the model from 200 Ma to present day. The velocity field as**
**denoted by the coloured arrows is generated by the plate tectonic reconstruction software GPlates based on EarthByte's 2013 plate**
**model (see text). This velocity field is used as a time-dependent velocity boundary condition for the TERRA model runs. The**
**model's temperature field within the lower mantle is shown in terms of the 2000 K and the 2800 K isotherm depicted as blue and**
**red isosurfaces, respectively.**



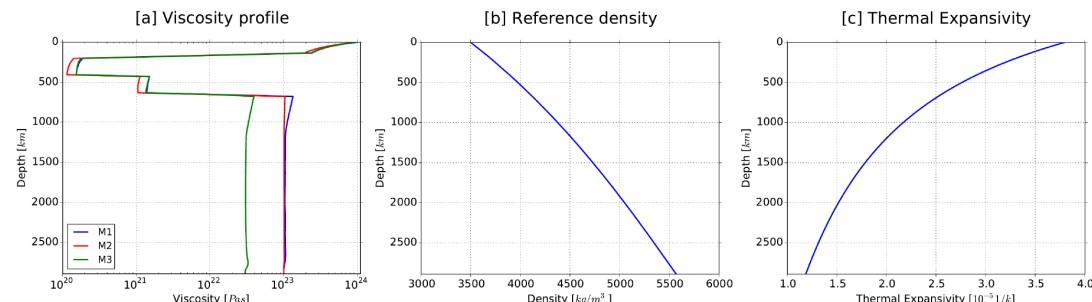

**Figure 2: Radial profiles of (a) viscosity, (b) density and (c) thermal expansivity of our models.**











**Figure 3: Dynamic topography time series of selected locations for all three models (M1-M3). Vertical motions are dominated by the plate tectonic history, while variations of core-mantle-boundary temperature (M2) and lower mantle viscosity (M3) have only marginal effects on dynamic topography. For locations of sample points, see Figs. 11 and 12.**

**Figure 4A: Evolution of global dynamic topographic elevation from 200 Ma to present. Blue colours denote negative dynamic topography, red colours mark positive dynamic topography. Figure A shows dynamic topography in the absolute reference frame. Indicated points show synthetic sample locations as discussed in the text. Note that sample points move along with their accordant tectonic plates. The evolution of dynamic topography in 10 Myr intervals can be found in the supplement.**





**Figure 4B shows dynamic topography in the plate frame of reference. The evolution of dynamic topography in 10 Myr intervals**
**can be found in the supplement.**






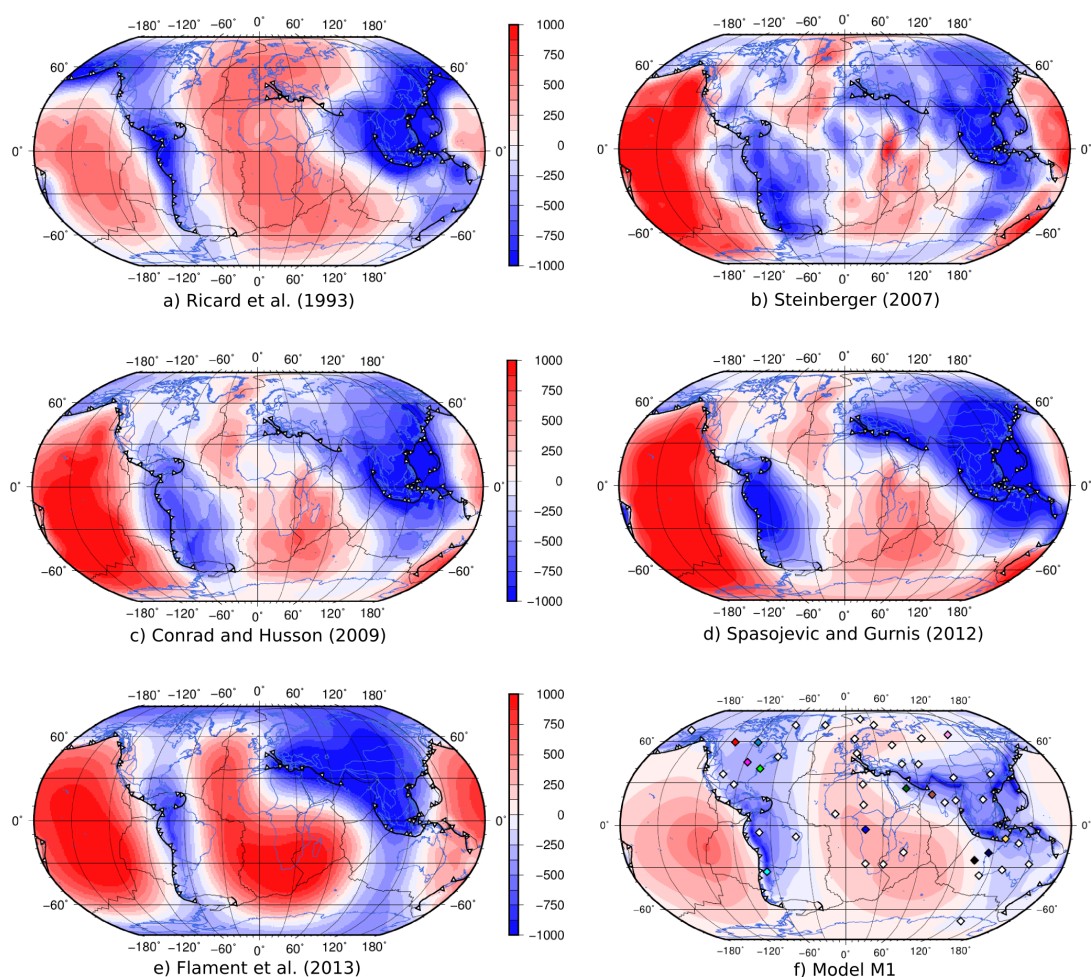

**Figure 5: Comparison of present-day global dynamic topography as inferred from published mantle convection models: a) Ricard et al. (1993); b) Steinberger (2007); c) Conrad and Husson (2009); d) Spasojevic and Gurnis (2012); e) Flament et al. (2013); f) our model M1. All grids display dynamic support that has been corrected for air-loading.**






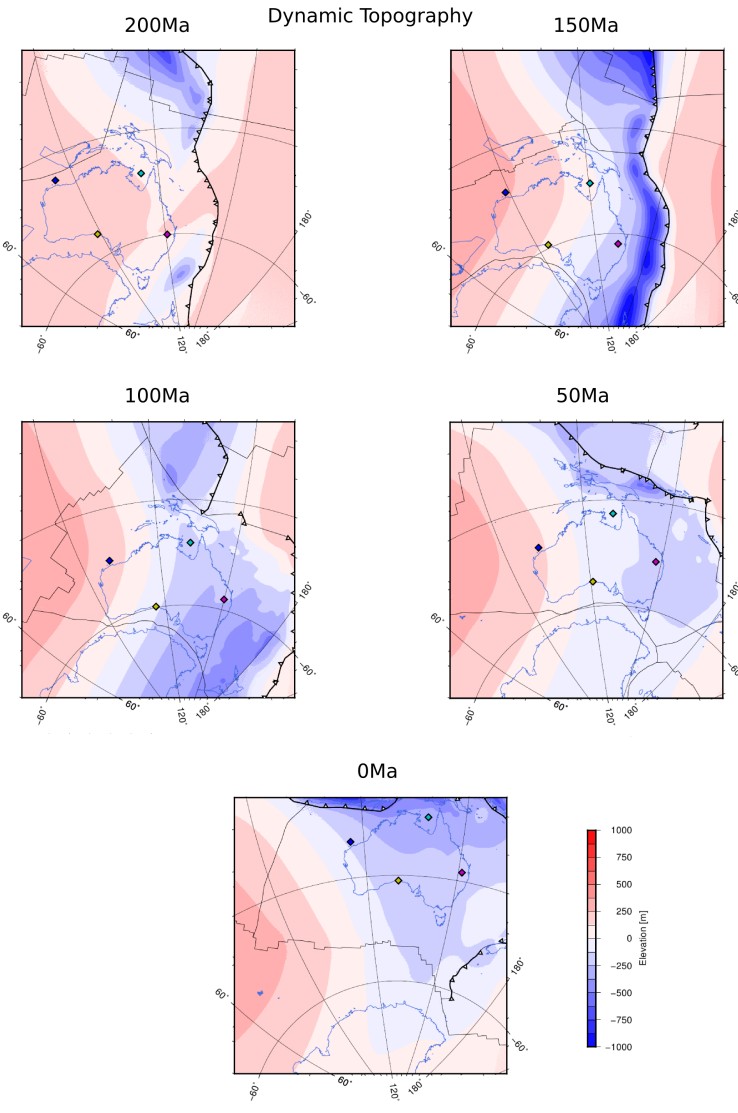


**Figure 6: Dynamic topography evolution for Australia, analogous to Fig. 4A. Between 200 Ma and 120 Ma Australia's eastern
margin experiences westward-directed subduction of the Phoenix plate. From 50 Ma onwards Australia's dynamic topography is
dominated by the north-ward directed subduction of the Australian plate under the Indonesian archipelago. See Supplementary
Material for dynamic topography evolution in 10 Myr time steps.**




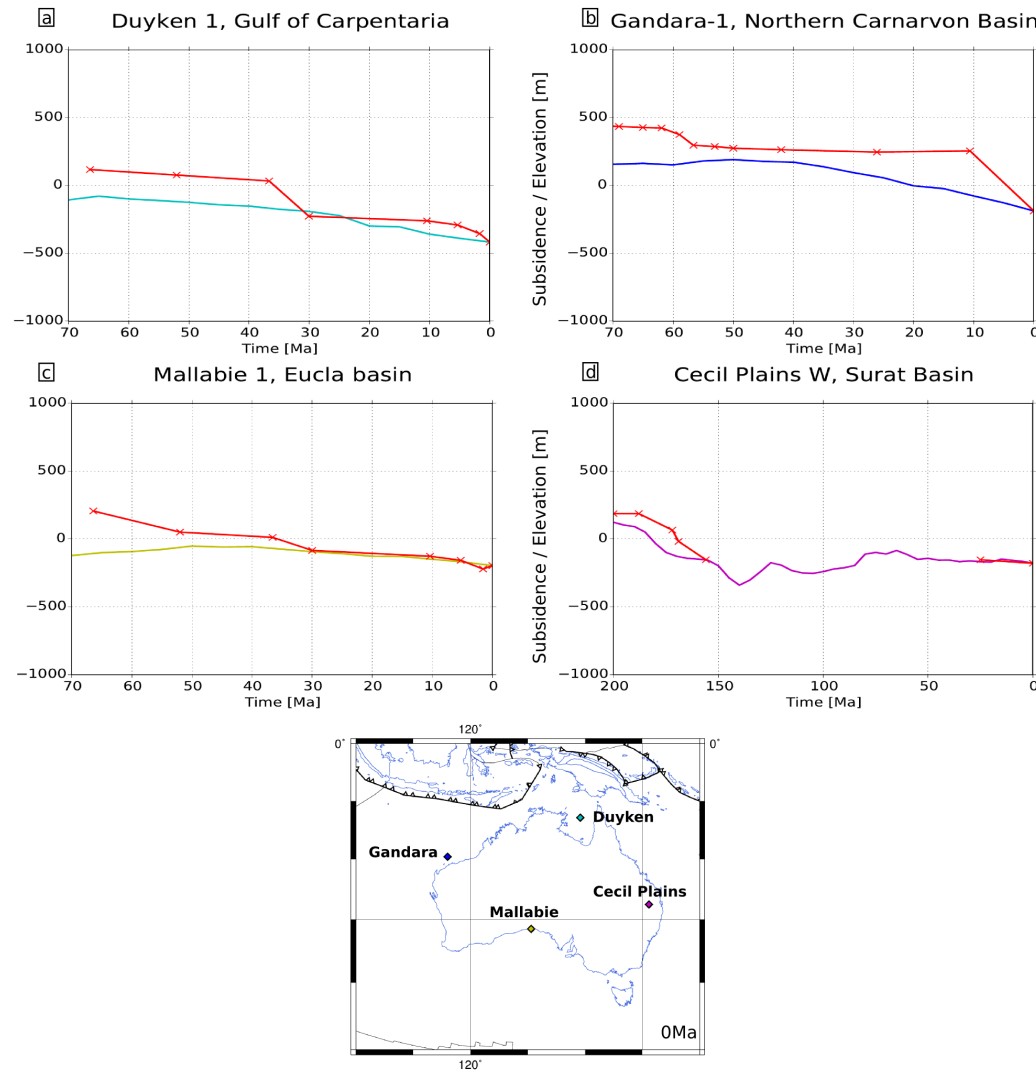

**Figure 7: Tectonic subsidence curves of selected wells in Australia (red) as compared to our model output (coloured). Well data**
**curves (accordingly coloured) are offset to model elevation at present-day.**





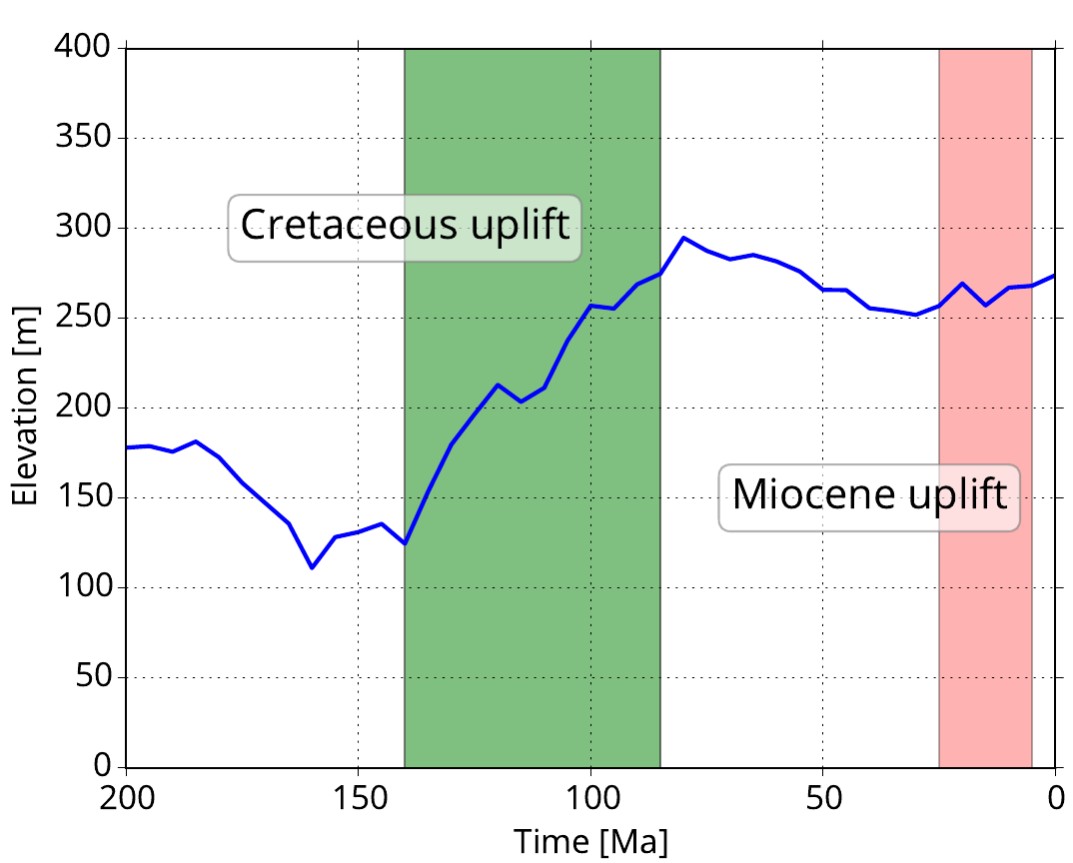

**Figure 8: South African (sample point South Africa East) uplift resulting from our models. While it shows the Cretaceous uplift as described in other works, the uplift in the Miocene period is not reproduced as the literature suggests (see text).**





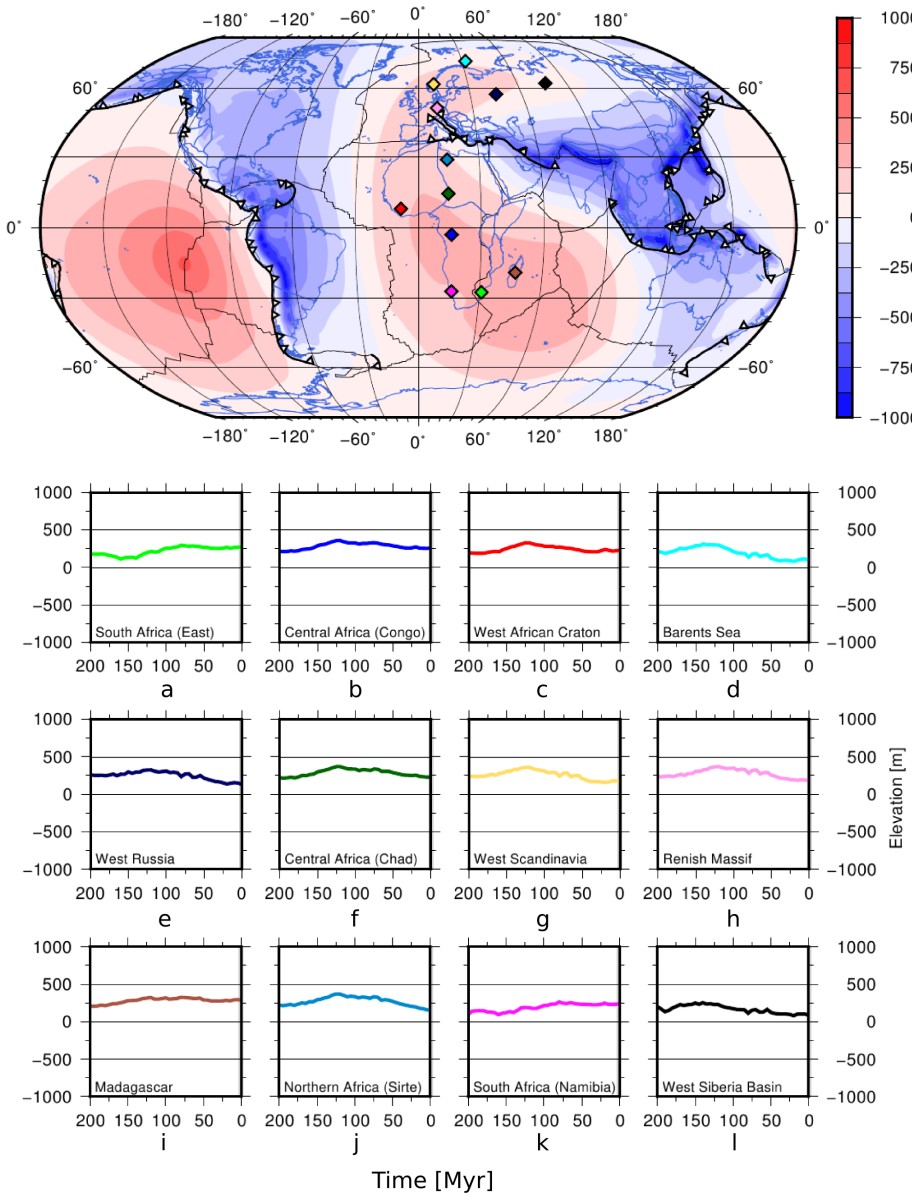

**Figure 9: Overview map (top) and time series (bottom) of dynamic topography for Category I, "Topographic stable areas".**





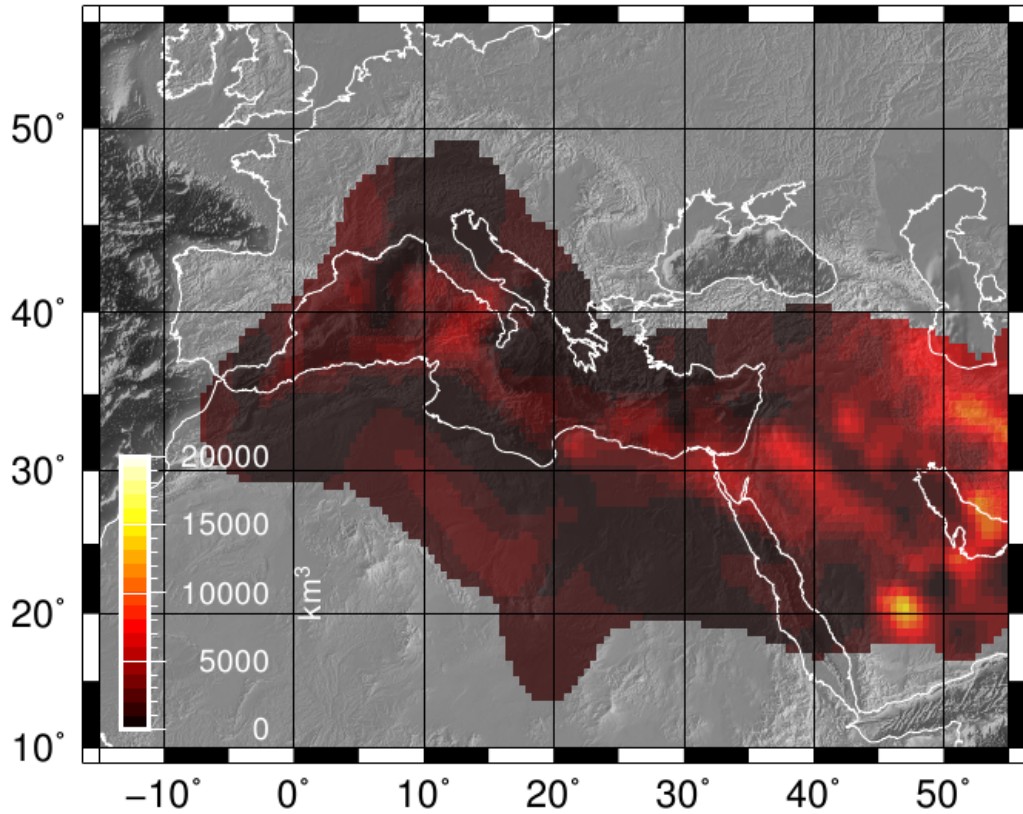

**Figure 10: Volume of subducted oceanic lithosphere (in km³) for the western Tethyan region for the past 200 Myr. The amount of**
**material was calculated based on paleo-age grids with 6 arc minute grid cell size for the subducted oceanic lithosphere (cf. Müller**
**et al. (2008) for methodology) and convergence velocities derived from the input plate kinematic model. We assume a simple half-**
**space cooling model to compute the thickness of the oceanic lithosphere for ages < 69.9 Myr as $z = 11.36\sqrt{age}$ where age is the**
**age of the oceanic lithosphere. Older lithosphere is set to a default thickness of 95 km thickness (Kanamori and Press, 1970).**
**Volumes are derived by multiplying thickness with computed trench-normal convergence in 1 Myr time steps. Grid cell size for**
**the subduction volume is 30 arc minutes. Map background is ETOPO1 topography (Amante and Eakins, 2009).**



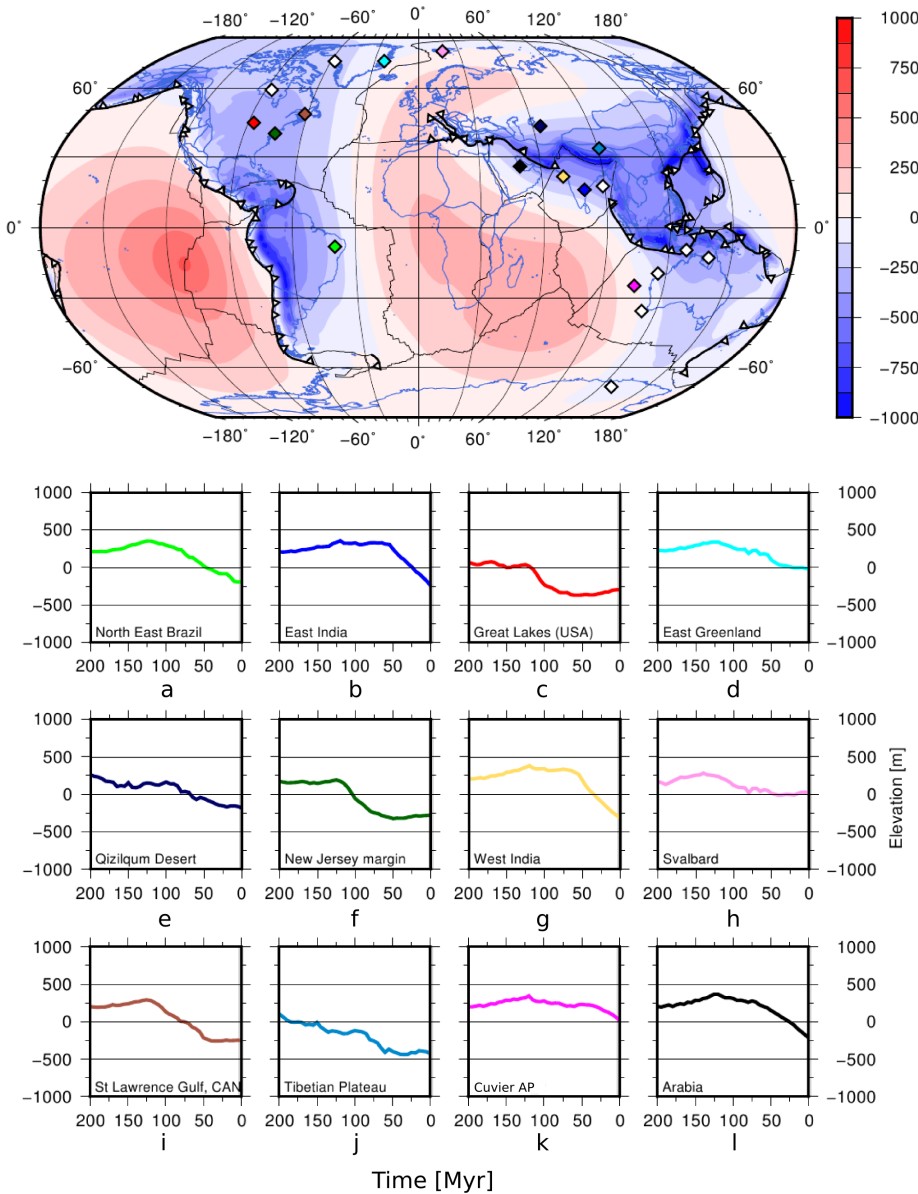

**Figure 11A: Overview (top) and time series (bottom) plot of dynamic topography for Category II, "Dynamically subsiding areas".**







**Figure 11B: Overview (top) and time series (bottom) plot of dynamic topography for Category II, "Dynamically subsiding areas", continued.**

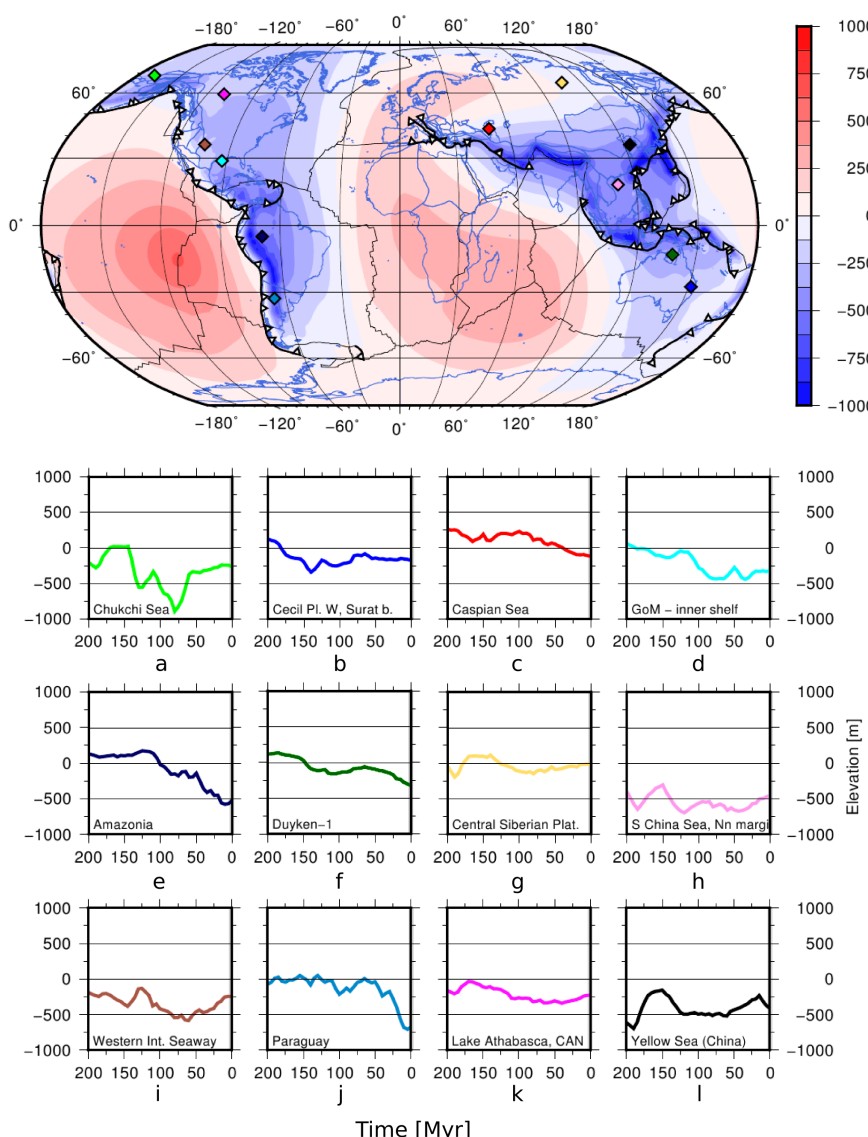

**Figure 12: Overview (top) and time series (bottom) plot of dynamic topography for Category III, "Fluctuating areas".**







**Figure 13: Comparison of dynamic topography time series at three sample locations with the relative orthogonal velocity of the down-going and overriding plate at the nearest point on the subduction zone. For locations of sample points, see Fig. 12.**



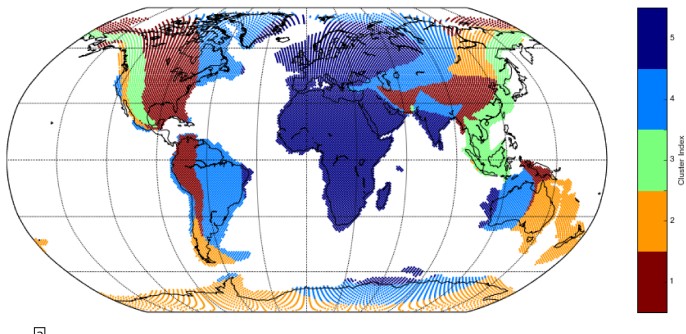

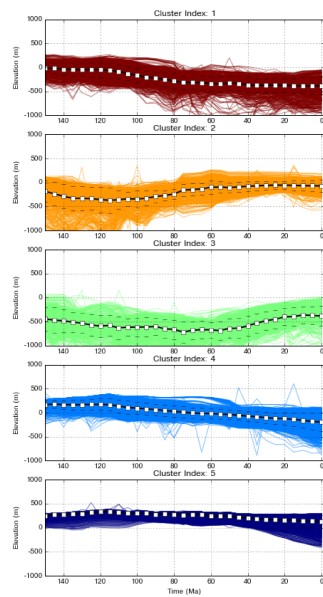

**Figure 14: The cluster analysis underpins the suggested categorisation; _k_ = 5. (a) The dark blue cluster consists of locations in stable areas, light blue corresponds to (long-term) dynamically subsiding areas, while the fluctuating category is encompassed by three clusters (green, orange and red). (b) Different clusters of elevation time series. The average trend is marked in black. See Supplementary Material for alternative choices of _k_.**





**Tables**

| Model parameters | |
| --- | --- |
| $T_{surf}$ (surface temperature) | 300 K |
| $T_{CMB}$ (CMB temperature) | (varied) |
| H (internal heating rate) | $5.5 \times 10^{-12}$ W kg$^{-1}$ |
| $\eta_{ref}$ (reference viscosity) | $1.0 \times 10^{20}$ Pa s |
| $\rho_S$ (surface density) | 3500 kg m$^{-3}$ |
| $\rho_{CMB}$ (CMB density) | 5568 kg m$^{-3}$ |
| $\alpha_S$ (surface thermal expansivity) | $3.8 \times 10^{-5}$ K$^{-1}$ |
| $\alpha_{CMB}$ (CMB thermal expansivity) | $1.2 \times 10^{-5}$ K$^{-1}$ |
| k (thermal conductivity) | 4.0 Wm$^{-1}$K$^{-1}$ |
| c (specific heat) | 1081.0 Jkg$^{-1}$K$^{-1}$ |

**Table 1: Physical parameters common to all model runs. See Table 2 for the parameters varied between models.**




| Model | Viscosity Structure (w.r.t $\eta_{ref}$) | | | | $T_{CMB}$ [K] |
|---|---|---|---|---|---|
| | L | UM | TZ | LM | |
| M1 | 1000 | 1 | 10 | 1000 | 4000 |
| M2 | 1000 | 1 | 10 | 1000 | 3500 |
| M3 | 1000 | 1 | 10 | 300 | 4000 |

**Table 2: Key parameters of models M1, M2, and M3. Only the viscosity profile (cf. Fig. 2) and the temperature at the core-mantle-**
**boundary (CMB) have been varied. The viscosity structure refers to a reference viscosity of $\eta_{ref}$ of $10^{20}$ Pa s and comprises**
**different viscosities for the following radial mantle layers: Lithosphere (L), upper mantle (UM), transition zone (TZ) and lower**
**mantle (LM).**





**Author contributions**

M.Rubey: conducting & analysing models;

S. Brune, R. Davies, D. Müller: model design;

C. Heine: subduction volume;

S. Williams: cluster analysis.

All authors contributed to interpreting results. The manuscript was written by M. Rubey with contributions from all authors.

**Competing interests**

The author declares, also on behalf of all co-authors, that no competing interests exist.

**Acknowledgements**

S. Brune was funded by the Marie Curie International Outgoing Fellowship 326115 and the Helmholtz Young Investigators Group CRYSTALS. C. Heine was supported by ARC Linkage Project LP0989312 with Shell E&P and TOTAL. D. R. Davies is funded by an ARC Future Fellowship (FT140101262). This research was supported by resources provided by the Pawsey Supercomputing Centre with funding from the Australian Government and the Government of Western Australia and with the assistance of resources from the National Computational Infrastructure (NCI), which is supported by the Australian Government. Leonardo Quevedo is acknowledged for the numerical routines to compute subduction volumes and the implementation of the GPlates TERRA output routines, along with John Cannon. The authors thank the employees of both super computing centres for their generous support and Geoscience Australia for their vast, open, easily accessible database.