# Peer review of "Global patterns of Earth's dynamic topography since the Jurassic"

_Solid Earth, 2017_

## Referee Comment (RC1) · Anonymous Referee #1 · 23 Apr 2017

In their paper "Global patterns of Earth's dynamic topography since the Jurassic", Rubey & co-authors used forward models of thermal convection in earth's mantle to infer time-dependent changes in dynamic (surface) topography over the last 200 million years. They also performed k-means clustering to address how the long-wavelength dynamic topography in continental regions is affected by subduction zone. While I am somewhat convinced that the cluster analysis is worthwhile publishing, I am very critical to other aspects (i.e., the methodology for modelling dynamic topography, the validity of thermal convection models, the absence of mantle plumes, etc.). I formed the impression that the study do not provide any advance in our understanding of flow-induced dynamic topography. Also, it is important that the limitations and hypothesis are more clearly presented. I think a major revision of the manuscript is needed before it can be accepted for publication. My comments are as follows.

[Figure]

1. The authors stated that their models are Earth-like in terms of surface heat flux (page 1, lines 10-15). However, they didn't provide a global value for the heat flux at the surface obtained by their simulations. What is more important, a successful forward model of mantle convection must satisfy the energy balance (e.g., Kellogg et al., Science (1999)). Authors didn't demonstrate this fundamental constraint for any thermal convection model, i.e., that the difference between the surface heat flux and the core-mantle boundary (CMB) flux is equal to the internal heating (22 TW according to Table 1?) over a period of 200 million years.

2. The authors imposed the past plate velocities, derived from a global plate kinematic model, as a surface boundary condition (page 3, lines 5-10). I understand this is one of modelling techniques for the surface boundary condition, but this is also a non-physical procedure - the mantle buoyancy should drive the plate motions not the other way around. The authors should discuss this.

3. Why did the authors apply "a 70 Myr long pre-conditioning stage"? Can they demonstrate that their simulations reach a (quasi-)steady state in 70 Myr?

4. In modelling dynamic topography, the authors employed an approach in which the effects of lithospheric buoyancy structure on dynamic topography are discarded. It is hard to comprehend this approach if we know that dynamic topography which arises from all density anomalies in the convecting mantle (including the lithosphere) provides the best fit to an independent, reference model of dynamic topography estimated by removing isostatically compensated crustal heterogeneity, described either by CRUST1.0 or CRUST2.0, from the observed present-day surface topography (see Forte et al. in Treatise on Geophysics (2015)).

5a. Additionally, a layer above 225 km depth encompass both lithosphere and asthenosphere (see e.g. the PREM model). The lithosphere and asthenosphere are closely coupled to the large-scale flow patterns generated much deeper in the mantle. Anyone who knows that the dominant control on the evolution of temperature anomalies in

the mantle is from advection - velocity x grad (temperature) - should understand that velocity is critical and this velocity (including in the shallow mantle) is produced by density anomalies at all depths in the mantle, and at long wavelengths this velocity thus depends on very deep density anomalies. So, one cannot "zero" out shallow density anomalies and pretend to model time-dependent convection correction.

5b. It is also enigmatic (no single explanation is provided by the authors) why did they use a constant value of 2150 K to replace the model's thermal structure above 225 km depth (and why this depth)? After all, what is the point of re-running the model with a 225 km thick layer of a constant temperature?! This approach in modelling dynamic topography is not Earth-like.

6. Why didn't the authors compare their predictions of the present-day dynamic topography to a crust-corrected dynamic topography which is an independent prediction? These independent reference models are also published by Forte et al. in Treatise on Geophysics (2015). Additionally, Glisovic & Forte, JGR (2016) have recently published the predictions of the present-day dynamic topography obtained by the forward integration of tomography-based, Cenozoic reconstructions.

7. It looks like their numerical models of thermal convection in earth's mantle do not produce a single plume (i.e., a hot upwelling) that rises through the mantle (see Figure 1). In other words, their model design and approach are not well suited to take into account the important effects of mantle plumes on dynamic topography (e.g., Moucha & Forte, Nature Geoscience (2011); Glisovic & Forte, Science (2017)). Therefore, the authors should clearly state that their predictions of the global pattern of Earth's dynamic topography since the Jurassic are solely subduction-driven. I even suggest to the authors to change the title in order to reflect this fact.

8. Furthermore, the absence of mantle plumes in simulations presented here implies that the CMB flux is very low. For example, the recent estimates of the CMB heat flux are about 15 TW (e.g., Pozzo et al., Nature (2012)). The low heat flux across

the CMB prevents the development of a strong thermal boundary at the bottom of the mantle (e.g., Stacey & Loper, Phys. Earth Planet. Inter. (1983)) which means that deep buoyancy forces associated with active, hot upwellings are largely dismissed in this study. Also, how well does the CMB heat flux fit into the energy balance?

9. Accordingly, the authors cannot define "'geodynamic rules' for how different surface tectonic settings are affected by mantle processes" (page 1, lines 15-20) if they employ numerical models of thermal convection that do not resolve all buoyancy forces (hot upwellings and cold downwellings) throughout the mantle. These 'rules' are also well-known relations, for example, the negative topography below the eastern and western margins of the Pacific has been already interpreted in terms of present-day and Ceno-zoic subduction history. Therefore, authors may choose to remove or tone down on this statement, as well as to cite related work.

10. I also strongly call into question the statements like this one: "Our models provide a predictive quantitative framework linking mantle convection with plate tectonics and sedimentary basin evolution, thus improving our understanding of how subduction and mantle convection affect the spatio-temporal evolution of basin architecture". As I mentioned above, thermal convection models are solely subduction-driven, thus they can address only subduction effects on continental regions (and this only to some extent, because models do not include topographic contributions from lithospheric buoyancy).

11. Rubey & co-authors stated, "In contrast to previous models, out model M1 agrees well with this suggestion, mapping deep mantle return flow above the Pacific and African LLSVPs at amplitudes not exceeding \pm1 km", but it seems they missed to explain that this finding is strongly biased by the model 'assumptions'. Namely, their result (Figure 5f) is very similar to dynamic topography predicted on the basis of density anomalies below 200 km depth given by Forte et al. in Treatise on Geophysics (2015) (see Figure 21b).

12. Perhaps, authors should consider comparing their results for Africa to those obtained by Moucha & Forte, Nature Geoscience (2011).

13. It seems 3D mesh is uniformly spaced (page 3, lines 15-20)? If this is true, then I am wondering, do the authors experience any numerical instability across the thermal boundary layers?

14. I appreciate the 3D view of model's temperature in Figure 1, but perhaps it would be more informative for a reader if the authors decide to show isotherms based on lateral temperature variations.

---

## Referee Comment (RC2) · Anonymous Referee #2 · 28 Apr 2017

Authors Rubey et al. presented a global geodynamic prediction of dynamic topography from 200 Ma to the present. They used a forward modeling approach by mostly considering the effect of sinking slabs. The results compare relatively well to other studies including those using inverse approached based on seismically converted present-day mantle density structures. I find the global framework of dynamic topography history provided in this study to be especially useful for further references in similar modeling work. I do have some minor suggestions for improvement (as discussed below). Overall, this work is solid and reasonable, and I would suggest publication with a minor/technical revision.

The topic of dynamic topography has generated enormous amount of discussion and debates recently. Part of this is due to different (sometimes erroneous) understanding on the principles of mantle dynamics, and most is due to the quantification of mantle

dynamic properties such as viscosity and density profiles of the mantle. The authors did a good job laying out this evolving discussion, especially the contrasting views on dynamic topography from numerical modeling and geopotential field studies. However, I think the paper could be further enhanced by providing more analyses on the differences between different dynamic models, especially over geographic regions where many debates exist. For example, the authors cited two earlier papers (e.g. Lithgow-Bertelloni and Silver, 1998; Gurnis et al., 2000) for African topography, but there works (especially the former) suggest much large dynamic uplift over south Africa, compared to the authors' result. A key reason, according to my understanding, is that these earlier works underestimated/neglected the compositional nature of the LLSVP below the region that has be progressively established since then (Isshi & Trump, 1999; Masters et al., 2000; Ni et al., 2002; McNamara & Zhong, 2005; Simmons et al., 2009; Liu & Zhong, 2016). I consider the current paper potential to serve as a cornerstone to establishing a framework on this important topic of dynamic topography, so some discussions like the over outline above would significantly strengthen this manuscript.

Minor comments: P 2, L5: Geoid is a result of deep density anomaly and/or surface topography. E.g., in theory, an isostatically balanced mass anomaly in the lower mantle sitting on the CMB may generate geoid without exiting dynamic topography. It is thus not entirely faire to use geoid as a direct constraint on dynamic topography.

P 3, top line: The cited work by (Glišović and Forte, 2017) is a quasi-reversibility method, instead of an adjoint method.

P 5, end of parag. 10: from the description, it is clearly that the presented model assumes no compositional anomaly for the LLSVPs on the CMB. While a forward model with a pure thermal mantle likely would predict similar dynamic topography to those inverse models (based on tomography) assuming a thermal-chemical origin of the LLSVPs, it would be worth clarifying on this, so that readers will not be confused/baffles by the apparently different modeling approaches but with similar results. This would be a good place to clarify on these existing confusions in the field.

P 7, L 10: The prediction over South Africa implies little absolute dynamic topography (∼200 meters), and even less dynamic uplift since the Mesozoic (Africa is a stable platform). This is indeed in contrast to earlier models that suggest large amplitude uplift (e.g., Lithgow-Bertelloni and Silver, 1998). The likely reason is that the earlier model assumes a pure thermal origin of the LLSVP and takes the geometry of the LLSVP from tomography, leading to a much larger dynamic uplift signal. Again, this should be discussed and clarified.

Final remark: I really like the cluster analysis in this paper, and this is partly why I think this work could be used as a framework on the concept of dynamic topography.

---

## Author Comment (AC1) · 31 May 2017

**Anonymous Referee #1:**

In their paper "Global patterns of Earth's dynamic topography since the Jurassic", Rubey & co-authors used forward models of thermal convection in earth's mantle to infer time-dependent changes in dynamic (surface) topography over the last 200 million years. They also performed k-means clustering to address how the long-wavelength dynamic topography in continental regions is affected by subduction zone. While I am somewhat convinced that the cluster analysis is worthwhile publishing, I am very critical to other aspects (i.e., the methodology for modelling dynamic topography, the validity of thermal convection models, the absence of mantle plumes, etc.). I formed the impression that the study does not provide any advance in our understanding of flow-induced dynamic topography. Also, it is important that the limitations and hypothesis are more clearly presented. I think a major revision of the manuscript is needed before it can be accepted for publication. My comments are as follows.

We thank Referee 1 for his thorough review. There seems to be a fundamental disagreement concerning the question which model approach is better suited to represent the long-term evolution of Earth's mantle. We hope that our explanations as well as an accordant amended version of our manuscript serve to clarify this.

1. The authors stated that their models are Earth-like in terms of surface heat flux (page 1, lines 10-15). However, they didn't provide a global value for the heat flux at the surface obtained by their simulations. What is more important, a successful forward model of mantle convection must satisfy the energy balance (e.g., Kellogg et al., Science (1999)). Authors didn't demonstrate this fundamental constraint for any thermal convection model, i.e., that the difference between the surface heat flux and the core-mantle boundary (CMB) flux is equal to the internal heating (22 TW according to Table 1?) over a period of 200 million years.

   Apologies for omitting this information from our original manuscript. We agree that satisfying Earth's energy balance is a fundamental point for any convection simulation. In the revised version, we discuss model heat flux and compare it to the observed surface heat flux in Section 2.1.

   We have added: "The present-day surface heat flux of model M1 is 34 TW, which is consistent with current estimates for Earth's surface heat flux (~47 TW) minus the contributions from crustal radioactivity (Davies & Davies, 2010). Of this 34 TW, ~30% is transferred across the CMB with the remainder introduced through internal heating."

2. The authors imposed the past plate velocities, derived from a global plate kinematic model, as a surface boundary condition (page 3, lines 5-10). I understand this is one of modelling techniques for the surface boundary condition, but this is also a non-physical procedure - the mantle buoyancy should drive the plate motions not the other way around. The authors should discuss this.

   The reviewer is correct that plate tectonics and mantle convection should evolve self-consistently. However, to our knowledge, there are no published models, to date, that capture the global dynamics of Earth's mantle and lithosphere self-consistently, particularly models which allow one to compare the geographic distribution of heterogeneity with the present day Earth. Owing to this, we opted for the approach used in our manuscript (i.e. prescribing plate motion histories as a kinematic boundary condition). This is an approach that is commonly utilised within the geodynamics community. The different approaches used to predict global dynamic topography are covered, at length, in the review paper of Flament et al. (2013) and a number of other studies. As such, we see no motivation for further discussing these approaches here.

3. Why did the authors apply "a 70 Myr long pre-conditioning stage"? Can they demonstrate that their simulations reach a (quasi-)steady state in 70 Myr?

Surface kinematic boundary conditions were fixed to those of the oldest available reconstruction for 70 Myr. This was to approximate the late Triassic mantle heterogeneity structure or, more specifically, late Triassic slab locations (i.e. the unknown initial condition). There was no requirement that the models reach a quasi-steady state in that time: indeed, allowing them to evolve towards a quasi steady state would result in large accumulations of subducted material at the mantle's base (below paleo subduction zones) which would have an overwhelming and unrealistic influence on the flow field (and hence our dynamic topography predictions).

4. In modelling dynamic topography, the authors employed an approach in which the effects of lithospheric buoyancy structure on dynamic topography are discarded. It is hard to comprehend this approach if we know that dynamic topography which arises from all density anomalies in the convecting mantle (including the lithosphere) provides the best fit to an independent, reference model of dynamic topography estimated by re- moving isostatically compensated crustal heterogeneity, described either by CRUST1.0 or CRUST2.0, from the observed present-day surface topography (see Forte et al. in Treatise on Geophysics (2015)).

There is a fundamental difference between the approach chosen in our study and in Forte et al. (2015, Treatise on Geophysics). Forte et al. build their model on mantle heterogeneities derived from seismic tomography. In our forward modelling approach, however, we generate mantle heterogeneities through time-dependent plate velocities at Earth's surface that are derived from a rigid plate reconstruction of the last 200 My. Our approach generates a self-consistent thermal boundary layer and therefore inserts slabs with age-dependent thickness into the model Earth, however, it does not reproduce continental lithospheric thicknesses. The reason is that the continental lithosphere thickness in nature is predominantly inherited from the Precambrian and, subsequently, modified by tectonic deformation, which are both beyond the scope of our approach. We therefore focus on long-wavelength, subduction-driven global flow below the lithosphere and in our dynamic topography computation we hence discard any stresses arising from lithospheric isostasy and lateral buoyancy variations within the lithosphere. This is done by blanking heterogeneities in the lithospheric depth range where our radial viscosity profile assigns a high viscosity of 1e23 Pas (i.e. in the upper 225 km). Note that similar approaches have been employed in various previous studies (e.g.: Lithgow-Bertelloni and Silver, 1998; Steinberger, 2007; Conrad and Husson, 2009; Flament et al. 2013, and Barnett-Moore et al. 2016.)

Please see point 5 for a clarification of this point within the manuscript.

5. Additionally, a layer above 225 km depth encompass both lithosphere and asthenosphere (see e.g. the PREM model). The lithosphere and asthenosphere are closely coupled to the large-scale flow patterns generated much deeper in the mantle. Anyone who knows that the dominant control on the evolution of temperature anomalies in the mantle is from advection - velocity x grad (temperature) - should understand that velocity is critical and this velocity (including in the shallow mantle) is produced by density anomalies at all depths in the mantle, and at long wavelengths this velocity thus depends on very deep density anomalies. So, one cannot "zero" out shallow density anomalies and pretend to model time-dependent convection correction.

As stated above, we focus on the long-wavelength, subduction-driven global flow component below the lithosphere as this is the contribution to dynamic topography that does not depend on specific knowledge about the structure of the lithosphere and the depth of the lithosphere asthenosphere boundary, as neither is known for the entire model period of 200 My. The depth above which we applied the blanking procedure has been chosen such that all lithospheric structures are encompassed. However we varied this limiting depth between 300 and 150 km

and the impact on the result was negligible. Interestingly, our results for present-day dynamic topography are very similar to the model of Forte et al. 2015 where only the deep-seated contribution of mantle flow is accounted for (below 200 km depth), as Referee 1 pointed out in comment #12.

We discuss this point in section 2.1 as follows: "Note that to calculate dynamic topography, we focus on long-wavelength subduction-driven global flow below the lithosphere as this is the contribution to dynamic topography that does not depend on specific structure of the lithosphere and the depth of the lithosphere asthenosphere boundary. To this aim, we replace the model's thermal structure above 225 km depth with the constant median mantle temperature of 2150 K. We subsequently re-run the model for one time step and use the resulting surface normal stresses to evaluate dynamic topography. The depth above which we applied the blanking procedure has been chosen such that all lithospheric structures are encompassed. However we varied this limiting depth between 300 and 150 km and the impact on the results were negligible. We also varied the blanking temperature of 2150 K, but since the chosen temperature is applied to the entire spherical shell, it is always neutral with respect to the mantle's buoyancy structure and the amplitude of dynamic topography does not depend on the specific temperature choice. Similar approaches to discard the lithospheric buoyancy structure from the dynamic topography signal have been applied by Lithgow-Bertelloni and Silver (1998), Steinberger (2007), Conrad and Husson (2009) and Flament et al. (2013) for structures above 325, 220, 300, and 350 km depth, respectively."

6. It is also enigmatic (no single explanation is provided by the authors) why did they use a constant value of 2150 K to replace the model's thermal structure above 225 km depth (and why this depth)? After all, what is the point of re-running the model with a 225 km thick layer of a constant temperature?! This approach in modelling dynamic topography is not Earth-like.

The temperature is arbitrarily set to the median mantle temperature (2150 K). However we tested a range of other temperatures. But since the chosen temperature is applied to the entire spherical shell, it is always neutral with respect to the mantle's buoyancy structure and the amplitude and wavelength of dynamic topography did not depend on the specific temperature chosen.

See point 5 for a clarification of this point within the manuscript.

7. Why didn't the authors compare their predictions of the present-day dynamic topography to a crust-corrected dynamic topography which is an independent prediction? These independent reference models are also published by Forte et al. in Treatise on Geophysics (2015). Additionally, Glisovic & Forte, JGR (2016) have recently published the predictions of the present-day dynamic topography obtained by the forward integration of tomography-based, Cenozoic reconstructions.

We compare our present-day results to 5 previous studies of which 3 models are based on tomography (Fig. 5 b,c,d). We hope the reviewer understands that for reasons of manuscript length we cannot compare our results to all models of dynamic topography that have ever been published.

8. It looks like their numerical models of thermal convection in earth's mantle do not produce a single plume (i.e., a hot upwelling) that rises through the mantle (see Figure 1). In other words, their model design and approach are not well suited to take into account the important effects of mantle plumes on dynamic topography (e.g., Moucha & Forte, Nature Geoscience (2011); Glisovic & Forte, Science (2017)). Therefore, the authors should clearly state that their predictions of the global pattern of Earth's dynamic topography since the Jurassic are solely subduction-driven. I even suggest to the authors to change the title in order to reflect this fact.

We opted to focus on the role of subducted slabs in controlling Earth's transient topographic expression. To this end, our models were configured in such a way that plumes play an insignificant role in the convective planform (over the time-scales of our model large plumes are only beginning to initiate at the CMB, whilst smaller plumes do not strongly modulate the flow regime). In models like ours, it is not possible to control/prescribe plume locations. As such, focussing on the first-order effects of subduction, which we could control, seemed like a reasonable first step. This also played a role in our decision to neglect density anomalies above 225 km depth, which are likely to induce shallow mantle flow regimes (e.g. edge-driven convection and shear-driven upwelling), which were not the focus of our study.

As suggested, we have modified the title of the manuscript to emphasize our focus on the role of subducted slabs: "Global patterns of Earth's dynamic topography since the Jurassic: the dominant role of subducted slabs"

9. Furthermore, the absence of mantle plumes in simulations presented here implies that the CMB flux is very low. For example, the recent estimates of the CMB heat flux are about 15 TW (e.g., Pozzo et al., Nature (2012)). The low heat flux across the CMB prevents the development of a strong thermal boundary at the bottom of the mantle (e.g., Stacey & Loper, Phys. Earth Planet. Inter. (1983)) which means that deep buoyancy forces associated with active, hot upwellings are largely dismissed in this study. Also, how well does the CMB heat flux fit into the energy balance?

Please see our response to point 1 above.

10. Accordingly, the authors cannot define "'geodynamic rules' for how different surface tectonic settings are affected by mantle processes" (page 1, lines 15-20) if they employ numerical models of thermal convection that do not resolve all buoyancy forces (hot upwellings and cold downwellings) throughout the mantle. These 'rules' are also well- known relations, for example, the negative topography below the eastern and western margins of the Pacific has been already interpreted in terms of present-day and Cenozoic subduction history. Therefore, authors may choose to remove or tone down on this statement, as well as to cite related work.

Once again we would like to emphasize that the study focuses on subduction-driven flow. Nonetheless, we did examine models with different (longer) initial conditions, where plumes were able to form. Comparisons show that, naturally, dynamic topography gets modified in some places due to the role of upwellings. However, these are shorter wavelength effects, which are much less significant than the role of subduction and, hence, our rules will generally (away from sites of active upwelling) remain applicable.

We added this clarification in section 3.1: "Nevertheless, we also examined models with different initial conditions, where plumes were able to form. We found that dynamic topography gets modified in regions of active upwelling. However, these dynamic topography highs constitute shorter wavelength features (Moucha & Forte, 2011; Hoggard et al., 2016) and, on a global scale, are less significant than the role of subducting slabs, which is why we do not examine plume-related dynamic topography in this study."

11. I also strongly call into question the statements like this one: "Our models provide a predictive quantitative framework linking mantle convection with plate tectonics and sedimentary basin evolution, thus improving our understanding of how subduction and mantle convection affect the spatio-temporal evolution of basin architecture". As I mentioned above, thermal convection models are solely subduction-driven, thus they can address only subduction effects on continental regions (and this only to some extent, because models do not include topographic contributions from lithospheric buoyancy).

As stated above we also investigated models with thermal buoyancy-driven plumes that introduced second-order effects on the global dynamic topography pattern, but these additional complexities did not change our conclusions on long-wavelength patterns. Furthermore it seems that the cited statement aligns very well with the viewpoint of Reviewer 2 who writes "I find the global framework of dynamic topography history provided in this study to be especially useful for further references in similar modelling work."

In addition, a significant number of publications over recent years have highlighted the influence of long-wavelength sub-lithospheric mantle convection on the evolution of continental topography and basin evolution (e.g. Salles et al., G3, 2017; Spasojevic & Gurnis, AAPG Bull, 2012; Flament et al., EPSL, 2014; Heine et al., Tectonophysics, 2010 ) which are corroborated by observational approaches (e.g. Czarnota et al., G-cubed, 2013).

12. Rubey & co-authors stated, "In contrast to previous models, out model M1 agrees well with this suggestion, mapping deep mantle return flow above the Pacific and African LLSVPs at amplitudes not exceeding pm1 km", but it seems they missed to explain that this finding is strongly biased by the model 'assumptions'. Namely, their result (Figure 5f) is very similar to dynamic topography predicted on the basis of density anomalies below 200 km depth given by Forte et al. in Treatise on Geophysics (2015) (see Figure 21b).

We thank Referee 1 for pointing this out. We added a reference to Forte et al. (2015) in the corresponding paragraph. That our model agrees with their results despite the different setup is an indication for the robustness of both approaches. As said above we explicitly focus on long-wavelength dynamic topography and we do not take into account lithospheric effects through our methodology.

The last part of section 3.2.1 now reads: "Other studies suggest that amplitudes for long-wavelength ($10^3$ km) dynamic topography should be smaller than ±1 km (Wheeler and White, 2002; Molnar et al., 2015; Forte et al., 2015; Hoggard et al., 2016). In contrast to some previous models, our model M1 agrees well with this suggestion, mapping deep mantle return flow above the Pacific and African LLSVPs at amplitudes not exceeding ±1 km."

13. Perhaps, authors should consider comparing their results for Africa to those obtained by Moucha & Forte, Nature Geoscience (2011).

The study of Moucha & Forte (2011) relates changes in East African and Arabian topography to the arrival of the Afar plume 30 Myr ago. Since our models do not focus on plume-related dynamic topography, the comparison to this study is unfortunately difficult.

14. It seems 3D mesh is uniformly spaced (page 3, lines 15-20)? If this is true, then I am wondering, do the authors experience any numerical instability across the thermal boundary layers?

The icosahedral discretisation of the spherical surface is almost uniform, although lateral spacing varies with radius. Our models have a lateral resolution of ~ 28 km at the surface and ~14 km at the CMB. Radial spacing is ~22 km across the entire domain, which ensures that our thermal boundary layers are well resolved. We have updated the manuscript to reflect this.

15. I appreciate the 3D view of model's temperature in Figure 1, but perhaps it would be more informative for a reader if the authors decide to show isotherms based on lateral temperature variations.

Figure 1 primarily serves to illustrate the degree of complexity in our mantle convection models and the time-dependency of the planforms. We believe that isotherms of absolute temperatures

are as well suited to accomplish this point as isotherms of lateral temperature variations and prefer not to modify Figure 1.

---

## Author Comment (AC2) · 31 May 2017

*Anonymous Referee #2:*

Authors Rubey et al. presented a global geodynamic prediction of dynamic topography from 200 Ma to the present. They used a forward modelling approach by mostly considering the effect of sinking slabs. The results compare relatively well to other studies including those using inverse approached based on seismically converted present-day mantle density structures. I find the global framework of dynamic topography history provided in this study to be especially useful for further references in similar modelling work. I do have some minor suggestions for improvement (as discussed below). Overall, this work is solid and reasonable, and I would suggest publication with a minor/technical revision. The topic of dynamic topography has generated enormous amount of discussion and debates recently. Part of this is due to different (sometimes erroneous) understanding on the principles of mantle dynamics, and most is due to the quantification of mantle dynamic properties such as viscosity and density profiles of the mantle.

The authors did a good job laying out this evolving discussion, especially the contrasting views on dynamic topography from numerical modelling and geopotential field studies. However, I think the paper could be further enhanced by providing more analyses on the differences between different dynamic models, especially over geographic regions where many debates exist. For example, the authors cited two earlier papers (e.g. Lithgow- Bertelloni and Silver, 1998; Gurnis et al., 2000) for African topography, but there works (especially the former) suggest much large dynamic uplift over south Africa, compared to the authors' result.

A key reason, according to my understanding, is that these earlier works underestimated/neglected the compositional nature of the LLSVP below the region that has be progressively established since then (Isshi & Trump, 1999; Masters et al., 2000; Ni et al., 2002; McNamara & Zhong, 2005; Simmons et al., 2009; Liu & Zhong, 2016).

I consider the current paper potential to serve as a cornerstone to establishing a framework on this important topic of dynamic topography, so some discussions like the over outline above would significantly strengthen this manuscript.

We are thankful for this motivating statement and accounted for all comments of Referee 2 below and in the revised version of the manuscript.

*Minor comments:*

1. P 2, L5: Geoid is a result of deep density anomaly and/or surface topography. E.g., in theory, an isostatically balanced mass anomaly in the lower mantle sitting on the CMB may generate geoid without exiting dynamic topography. It is thus not entirely fair to use geoid as a direct constraint on dynamic topography.

   We agree and removed this statement.

   The sentence in section 1 now reads: "Dynamic topography generates significant surface deflections in both continental and oceanic regions (e.g. Ricard et al., 2006)."

2. P 3, top line: The cited work by (Glišovic and Forte, 2017) is a quasi-reversibility method, instead of an adjoint method.

   Corrected.

The revised sentence in section 1 is: "These studies examined the temporal evolution of dynamic support by forward modelling (e.g. Ricard et al., 1993; Zhang et al., 2012; Flament et al., 2013), backward advection (e.g. Conrad and Gurnis, 2003; Moucha et al., 2008; Heine et al., 2010), adjoint schemes (e.g. Bunge et al., 2003; Liu et al., 2008; Spasojevic et al., 2009; Colli et al. 2017), or a back-and-forth iterative method (Glišović and Forte, 2016, 2017)."

3. P 5, end of parag. 10: from the description, it is clearly that the presented model assumes no compositional anomaly for the LLSVPs on the CMB. While a forward model with a pure thermal mantle likely would predict similar dynamic topography to those inverse models (based on tomography) assuming a thermal-chemical origin of the LLSVPs, it would be worth clarifying on this, so that readers will not be confused/baffles by the apparently different modelling approaches but with similar results. This would be a good place to clarify on these existing confusions in the field.

We agree that there is an ongoing controversy regarding the thermo-chemical nature of LLSVPs and in the revised version we explicitly clarified that we are using a purely thermal model without any compositional heterogeneities.

We added this clarification in section 2.1: "Note that for simplicity our models assume no chemical heterogeneity. The distribution of heterogeneity is solely controlled by the imposed plate motion histories, the material properties of the model and the heating mode. Purely thermal models as in the present study have previously been shown to provide a good match to seismic observations (Schuberth et al. 2009; Davies et al. 2012; 2015) and dynamic topography (Colli et al. 2017)."

4. P 7, L 10: The prediction over South Africa implies little absolute dynamic topography (~200 meters), and even less dynamic uplift since the Mesozoic (Africa is a stable platform). This is indeed in contrast to earlier models that suggest large amplitude uplift (e.g., Lithgow-Bertelloni and Silver, 1998). The likely reason is that the earlier model assumes a pure thermal origin of the LLSVP and takes the geometry of the LLSVP from tomography, leading to a much larger dynamic uplift signal. Again, this should be discussed and clarified.

The models of Lithgow-Bertelloni and Silver are very different to ours. They are instantaneous in nature, have a simplified depth-dependent rheology and the buoyancy field is derived from tomography (with seismic anomalies converted to density anomalies using a constant scaling factor which is oversimplified). Our model also assumes a purely thermal origin for heterogeneity, but the distribution of heterogeneity is controlled by plate motion histories (see point above). Furthermore, we incorporate a temperature dependent viscosity and also account for mantle compressibility, which will both lead to differences. We emphasize that whilst our models do yield different predictions to the earliest models by Lithgow-Bertelloni and Silver, they are generally consistent with more recent models by, for example, Flament et al. (2013).

5. Final remark: I really like the cluster analysis in this paper, and this is partly why I think this work could be used as a framework on the concept of dynamic topography.

Once again we would like to thank Referee 2 for his/her constructive comments.